# Longitudinal Association between Uric Acid and Incident Sarcopenia

**DOI:** 10.3390/nu15143097

**Published:** 2023-07-11

**Authors:** Shengliang Zhou, Limin Wu, Haibo Si, Bin Shen

**Affiliations:** Department of Orthopedic Surgery and Orthopedic Research Institute, West China Hospital, Sichuan University, Chengdu 610041, China; zhoushengliang@stu.scu.edu.cn (S.Z.); wulimin@stu.scu.edu.cn (L.W.); sihaibo@schscu.cn (H.S.)

**Keywords:** sarcopenia, uric acid, longitudinal study, CHARLS

## Abstract

Background: Sarcopenia has emerged as a significant public health concern. Uric acid (UA), as a metabolite with excellent antioxidant capacity, has been found to be associated with sarcopenia. However, the casual effects of UA on incident sarcopenia still remain unclear. Our study aimed to explore the longitudinal association between UA and incident sarcopenia among middle-aged and older adults. Method: A total of 5086 participants aged ≥45 years old without sarcopenia at baseline were included from the China Health and Retirement Longitudinal Study (CHARLS). Due to the sex differences, the UA levels were analyzed by categorizing into sex-specific quartiles or by using UA levels as a continuous variable (per 1 mg/dL). The longitudinal association between UA and incident sarcopenia was evaluated using Cox proportional hazards regression models. Results: During the 4-year follow-up period, 552 (10.85%) participants with incident sarcopenia were identified, of which 370 cases were males and 182 cases were females. Compared to the first quartile (Q1) UA levels, the Q3 and Q4 UA levels were significantly associated with lower risk of incident sarcopenia in males (Q3: adjusted hazard ratio (HR), 0.72; 95% CI (confidence interval), 0.54–0.97; Q4: HR, 0.57; 95% CI, 0.41–0.80). When UA was as a continuous variable (per 1 mg/dL), the association in males remained significant (HR: 0.87; 95% CI: 0.79–0.97). No significant association was observed in females. Conclusion: Our findings indicated that UA was negatively associated with incident sarcopenia in males but not in females among middle-aged and older Chinese.

## 1. Introduction

Sarcopenia is a generalized and progressive disease characterized by the accelerated loss of muscle mass and function, which could increase the risk of various adverse outcomes such as falls, frailty, and even mortality [1]. Sarcopenia is commonly associated with the aging process. However, it is important to acknowledge that lifestyle factors and chronic diseases also contribute significantly to the development and progression of sarcopenia [2,3]. The prevalence of sarcopenia worldwide exhibits considerable variation, ranging from 10% to 27%, due to the diverse diagnostic criteria and targeted population [4]. As the global population continues to age, the burden of sarcopenia is expected to rise substantially. Thus, identifying the modifiable risk factors that contribute to the development and progression of sarcopenia is crucial to prevent or slow down the progression of sarcopenia. Physical inactivity and poor diet have been reported to be risk factors of sarcopenia, but the association between uric acid and sarcopenia remained unclear [1]. 

Uric acid (UA) is a metabolite produced through the breakdown of purines in the body, which has excellent antioxidant capacity [5]. UA levels have been linked to various diseases including hypertension, cardiovascular disease, and chronic kidney disease in many studies [6,7]. According to previous studies, UA was reported to be positively associated with muscle mass in Brazilian [8] and Chinese [9] participants and positively associated with muscle strength in Japanese [10], Italian [11], and American [12] participants. Conversely, there have also been studies reporting that higher UA levels were associated with lower muscle mass [13] and weaker muscle strength [14]. Most of the studies focused on investigating the effects of UA on muscle mass and strength. Only one cross-sectional study performed in China has specifically explored the association between UA and sarcopenia, revealing that higher UA levels were associated with decreased risk of sarcopenia [15]. Although previous studies have found the association between UA and sarcopenia, the casual effects of UA on incident sarcopenia were still unclear. Therefore, to better understand the effects of UA on sarcopenia, we conducted this study to explore the longitudinal association between UA and incident sarcopenia among the middle-aged and older adults based on the nationally representative data from the China Health and Retirement Longitudinal Study (CHARLS).

## 2. Methods

### 2.1. Study Population

The study population was derived from the CHARLS, an ongoing nationally representative longitudinal survey of individuals aged 45 years and above in China. The CHARLS employed a multistage probability proportional to size sampling strategy to select households from 450 villages or residential communities in 28 provinces throughout China All recruited participants were interviewed through standardized face-to-face computer-assisted personal interview (CAPI) by well-trained interviewers. The baseline survey was conducted between June 2011 and March 2012, with a total of 17,708 participants being included. The follow-up survey was conducted every two years. A detailed description of the CHARLS design has been reported previously [16].

In this longitudinal study, we used the data from CHARLS 2011 (wave 1), 2013 (wave 2) and 2015 (wave 3). In wave 1, the sample size was 17,708. Participants lacking the information of sarcopenia (*n* = 4635), aged <45 or missing age information (*n* = 258), missing sex information (*n* = 9) and lacking the data of UA (*n* = 3550) were excluded. A total of 9256 participants had complete information at wave1. We further excluded participants with sarcopenia at baseline (wave 1, *n* = 1008) and participants missing information in the follow-ups (*n* = 3162). Finally, 5086 participants were included in the final analysis (Figure 1). 

Ethical approval for the CHARLS was obtained from the Institutional Review Board of Peking University (approval number: IRB00001052–11015), and written informed consents were obtained from all participants prior to enrollment in this survey [16]. This study was conducted following the Strengthening the Reporting of Observational Studies in Epidemiology (STROBE) reporting guidelines.

### 2.2. Diagnose of Sarcopenia

According to the 2019 consensus of the Asian Working Group for Sarcopenia (AWGS), sarcopenia diagnosis was assessed upon three parameters: muscle strength, appendicular skeletal muscle mass (ASM), and physical performance [17]. Participants with low muscle mass plus low muscle strength or low physical performance was diagnosed sarcopenia. Muscle strength was assessed by handgrip strength and the method of handgrip strength measurement has been previously documented [16]. Low muscle strength is defined as an average handgrip strength <28 kg for male and <18 kg for female, as per the aforementioned AWGS 2019. According to the established literature, the appendicular skeletal muscle mass (ASM) in the present study was estimated using a validated formula previously reported in the Chinese population, which has shown good consistency with dual energy X-ray absorptiometry (DXA) in various studies [18,19]. Specifically, the formula for ASM calculation was as follows: ASM = 0.193 × weight (kg) + 0.107 × height (cm) − 4.157 × sex − 0.037 × age (years) − 2.631 [18]. The body weight and height were measured utilizing the OmronTM HN-286 scale and SecaTM213 stadiometer, respectively. The male was encoded as 1 and female as 2. The skeletal muscle mass index (SMI) was computed by dividing ASM by the square of height (ASM/height^2^). Based on previous studies, the criterion of low muscle mass was set as the sex-specific lowest 20% of SMI in the study population, which corresponded to a value of <6.99 kg/m^2^ in male and <5.27 kg/m^2^ in female [19,20]. Physical performance was assessed by gait speed and 5-time chair stand test and low physical performance was defined as the gait speed <1.0 m/s or the 5-time chair stand test ≥12 s [17]. 

### 2.3. Uric Acid

All participants were asked to fast overnight and the blood samples of participants were collected by medically-trained staff from Chinese center for disease control and prevention (Chinese CDC), based on a standard protocol. Venous blood was separated into plasma and buffy coat and stored in cryovials. These cryovials were promptly frozen at −20 °C and transported to the Chinese CDC in Beijing within two weeks. Upon arrival, the samples were transferred to a deep freezer and stored at −80 °C until they were assayed at the laboratory of Capital Medical University. UA levels (mg/dL) were quantified using the UA Plus method [16]. 

### 2.4. Covariates

The following baseline variables were selected as covariates in this study, including demographic, health behaviors, chronic diseases, and biochemical parameters. The baseline demographic variables included age, sex, residence (rural and urban), highest educational level (primary school or below, middle school and high school or above), and marital status (married and living with spouse and others). Health behaviors included drinking status (yes and no), smoking status (yes and no) and self-reported nighttime sleep duration. Diabetes mellitus (DM) was diagnosed when participants self-reported having DM diagnosed by a physician or the fasting glucose levels ≥7 mmol/L or the random glucose levels ≥11.1 mmol/L or the glycosylated hemoglobin (HbA1c) ≥6.5%. Hypertension was defined as self-reported having hypertension diagnosed by a physician or the systolic blood pressure (BP) ≥140 mmHg or the diastolic BP ≥90 mmHg. Depression was assessed by the 10-item Center for Epidemiologic Studies Depression Scale (CES-D), and participants with scores ≥12 were considered to have depression. Dyslipidemia was defined as the ratio of cholesterol and high-density lipoprotein cholesterol (HDL-c) of ≥5.0 or self-reported having dyslipidemia diagnosed by a physician. Other chronic diseases such as pulmonary disease, kidney disease, cardiovascular disease, and arthritis were assessed by the physician-diagnosed history. The estimated glomerular filtration rate (eGFR) was calculated using creatinine values, age and sex [21]. 

### 2.5. Statistical Analysis

The sex differences in UA are well reported in several studies [22,23]. Therefore, all analyses were performed separately for males and females. The association between UA and sarcopenia was analyzed by categorizing UA levels into sex-specific quartiles or by using UA as a continuous variable (per 1 mg/dL). The first quartile of UA level (Q1) was set as the reference group (Male: Q1: ≤4.06 mg/dL, Q2: 4.06–4.80 mg/dL, Q3: 4.80–5.67 mg/dL, Q4: ≥5.67 mg/dL; Female: Q1: ≤3.30 mg/dL, Q2: 3.30–3.88 mg/dL, Q3: 3.88–4.58 mg/dL, Q4: ≥4.58 mg/dL). 

The baseline characteristics of including participants were described and compared according to the sex-specific quartiles of UA. Continuous variables and categorical variables were presented as means with standard deviations (SDs) and numbers with percentages, respectively. The characteristic differences between quartiles groups were compared using one-way analysis of variance for continuous variables and chi-square test for categorical variables. To assess the association between UA and sarcopenia in this four-year follow-up study, we employed Cox proportional hazards models to estimate the hazard ratios (HRs) with corresponding 95% confidence intervals (CIs). The proportional hazard assumption in the Cox proportional hazards models was verified using Schoenfeld residuals [24]. The fully multivariable model adjusted for age, residence, educational level, marital status, drinking status, smoking status, nighttime sleep duration, DM, hypertension, depression, dyslipidemia, pulmonary disease, kidney disease, cardiovascular disease, arthritis, eGFR, high sensitivity C-reactive protein (CRP), HDL-c and low-density lipoprotein cholesterol (LDL-c). Subgroup analyses stratified by age (median age), residence, smoking, drinking, hypertension, DM, dyslipidemia, and depression were performed.

Several sensitivity analyses were conducted to exam the robustness of the findings. First, we compared the baseline characteristics of included and excluded participants to explore the potential selection bias. Second, participants with hyperuricemia were excluded. Hyperuricemia was defined as baseline plasma UA > 7.0 mg/dL in males and >6.0 mg/dL in females [25]. Third, participants with eGFR ≤ 60 mL/min per 1.73 m^2^ were excluded. 

All statistical analyses in this study were conducted by STATA/MP, version 17.0. The level of statistical significance was set at *p* < 0.05 (two-sided).

## 3. Results

### 3.1. Baseline Characteristics Based on the UA Quartiles

A total of 5086 participants were included in this study. Of these, 2359 (46.38%) participants were males and 2727 (53.62%) participants were females. The baseline characteristics of the total sample based on the sex were shown in Table 1. UA levels were significantly higher in males (4.93 ± 1.23) than females (4.00 ± 1.05) at baseline (Table 1). Table 2 showed the baseline characteristics of males according to the quartiles of UA. Compared to males with lower UA levels, males with higher UA were more likely living in urban, to be non-smokers, and more likely having hypertension, depression, dyslipidemia, as well as having higher creatinine, LDL-c, CRP and lower eGFR and HDL-c (Table 2). Males with higher UA levels had higher SMI and better 5-time chair stand test performance. The incidence of sarcopenia increased across UA quartiles in males (Table 2). The baseline characteristics of females based on the quartiles of UA were shown in Table 3. Females with higher UA levels were older, and more likely living in urban, having DM, hypertension, dyslipidemia, arthritis, and higher creatinine, LDL-c, CRP, and lower eGFR and HDL-c. In female participants, those with elevated levels of UA exhibited higher SMI and no statistically significant difference in the incidence of sarcopenia was observed across the quartiles of UA levels.

### 3.2. Association between UA and Incident Sarcopenia 

During the 4 years follow-up, a total of 552 (10.85%) cases with incident sarcopenia were identified, of which 370 cases were males and 182 cases were females. Among males, the incidence rate of sarcopenia was 60.13 per 1000 person-years in Q1 UA group, 44.24 per 1000 person-years in Q2 UA group, 38.18 per 1000 person-years in Q3 UA group and 28.27 per 1000 person-years in Q4 UA group. Table 4 presented the HRs and 95% CIs for incident sarcopenia based on the sex-specific quartiles of UA and continuous variable UA (per 1 mg/dL). In males, model 1, which adjusted for age, residence, educational level and marital status demonstrated that the HRs and 95% CI for incident of sarcopenia in the Q2, Q3 and Q4 groups of UA compared to the Q1 group were 0.73 (0.56–0.95), 0.62 (0.47–0.82), 0.44 (0.32–0.60), respectively. The association did not significantly change after further adjusting for smoking, drinking, and nighttime sleep duration (model 2). Additionally, after further adjusting for DM, hypertension, depression, dyslipidemia, pulmonary disease, kidney disease, cardiovascular disease and arthritis (model 3) and further adjusting for eGFR, CRP, HDL-c and LDL-c based on the model 3 (model 4), males with Q3 and Q4 groups of UA were significantly associated with decreased risk of incident sarcopenia [model 3, Q3: 0.73 (0.55–0.97), Q4: 0.56 (0.41–0.76); model 4, Q3: 0.72 (0.54–0.97), Q4: 0.57 (0.41–0.80)]. When the UA was as a continuous variable (per 1 mg/dL), UA levels was negatively associated with the risk of incident sarcopenia in males. The HR and 95% CI in model 4 was 0.87 (0.79–0.97) (Figure 2A) (Table 4). 

In females, the incidence rate of sarcopenia in Q1–Q4 groups were 17.26, 18.46, 16.48 and 16.21 per 1000 person-years, respectively. There was no significant longitudinal association between UA and incident sarcopenia in females according to the quartiles of UA or as a continuous variable (per 1 mg/dL) (Figure 2B) (Table 4). Subgroup analyses stratified by median age, residence, smoking, drinking, hypertension, DM, dyslipidemia, and depression were conducted based on separate analyses of sex (Table 5). The association between UA as a continuous variable and incident sarcopenia was assessed in the subgroup analyses. 

According to the subgroup analyses, males who were younger (≤58 years old), living in rural, smokers, non-drinkers, and without hypertension, DM, dyslipidemia, as well as with depression were more responsive to the protective effects of UA against the incident of sarcopenia (Table 5). In females, there was no significant effect modification observed in subgroups analyses.

### 3.3. Sensitivity Analysis

We compared the baseline characteristic between included and excluded participants and the difference was not significant (Appendix A). After excluding participants with hyperuricemia and eGFR ≤60 mL/min per 1.73 m^2^, we reanalyzed the data and found that the association between UA and sarcopenia did not change significantly (Appendix A).

## 4. Discussion

We conducted a longitudinal study using nationally representative data to assess the association between UA and incident sarcopenia, diagnosed using AWGS 2019 algorithm, in middle-aged and older Chinese population. When the UA was analyzed by sex-specific quartiles, the Q3 and Q4 groups in males were significantly associated with decreased risk of incident sarcopenia. This negative association remained significant when considering UA as a continuous variable in males. In our analysis of females, we did not find a significant association between UA and sarcopenia. 

Our findings indicated a significant negative association between UA and incident sarcopenia in males but not in females. The potential protective effect of UA against sarcopenia are in line with previous studies. A cross-sectional study demonstrated that UA levels were positively associated with muscle mass and muscle strength in kidney transplant patients [8]. Another population-based study revealed that UA was positively associated with peak muscle strength [12]. In addition, a prospective study with 3-year follow-up reported that higher UA levels were associated with higher follow-up muscle strength after adjusting for potential confounders [26]. However, the sex difference of our findings showed inconsistences with previous studies. UA was found to be positively associated with muscle strength in older men and women in a cross-sectional study [12]. Another cross-sectional study conducted in West China reported that evaluated UA levels were negatively associated with sarcopenia in both males and females [15]. The results disparity may be attributed that participants in the study were all from four provinces in China (Yunnan, Guizhou, Sichuan, and Xinjiang), and that most of them were middle-aged ranging from 50 to 60 years old [15]. Nevertheless, there was also a retrospective cohort study supporting our findings [27]. This study revealed that the UA was positively associated with relative ASM in patients with peritoneal dialysis and this association only existed in males [27]. However, the study population was limited to peritoneal dialysis patients, and further studies involving the general population are needed. Compared with previous cross-sectional studies, our findings provided new evidence regarding the casual effects of UA on incident sarcopenia.

The negative association between UA and incident sarcopenia may be related with the excellent antioxidant capacity of UA. It has been well established that oxidative stress was an essential pathogenetic process in age-related sarcopenia, which was accompanied by the overproduction of reactive oxygen species [1,28]. UA may protect the muscle health through scavenging reactive oxygen species and reducing oxidative damage [29]. Moreover, UA is mainly derived from the metabolism of purine-rich foods, such as meats and seafood, which are rich in protein. Higher UA levels may reflect better nutritional status, which may partially explain the association between UA and sarcopenia [30]. The exact mechanisms of the effects of UA on sarcopenia were still unclear, and further studies are needed to explore its underlying mechanisms. 

For the sex difference of the association between UA and incident sarcopenia in our study, it may be related with the hormonal influence. Most females included in this study were menopausal, which resulted in a decrease of estrogen. Estrogen has been reported to promote the excretion of UA, which may result in elevated UA levels in menopausal females [31]. In addition, the decrease of estrogen could increase pro-inflammatory cytokines, such as tumor necrosis factor alpha or interleukine-6, which contributes to muscle catabolism [32,33]. Alterations in estrogen levels before and after menopause may partially attenuate the significant association between UA and incident sarcopenia in females. Moreover, the decrease of age-related muscle mass was more apparent in males, which may contribute to a more discernible protective effect of UA in males [34]. 

The strength of this study was based on a large and nationally representative sample, which allows for the extension of the findings to the middle-aged and older population in China. To our knowledge, this is the first study to explore the longitudinal association between UA and incident sarcopenia, assessed using the AWGS 2019 algorithm. However, there are also several limitations should be noted. First, although the formula has been previously validated in Chinese population and showed good consistency with DXA in various studies, the muscle mass was estimated by a formula, not by DXA or bioelectrical impedance analysis [18]. Second, in this study, UA was measured only once time. Although the assays were all preformed at Capital Medical University laboratory which had the same quality control, the impact of dynamic changes in UA levels on sarcopenia was unknown [16]. Third, we have adjusted multiple confounders, but other unmeasured confounders were not included, such as nutritional status and dietary intake. Finally, participants in this study were all from China and therefore the generalizability of our findings to other population groups may be limited.

## 5. Conclusions

In conclusion, our findings indicated that UA was negatively associated with incident sarcopenia in males but not in females in middle-aged and older Chinese. The findings supported the association between UA and sarcopenia, and suggested the gender difference existed in the effects of UA on sarcopenia.

## Figures and Tables

**Figure 1 nutrients-15-03097-f001:**
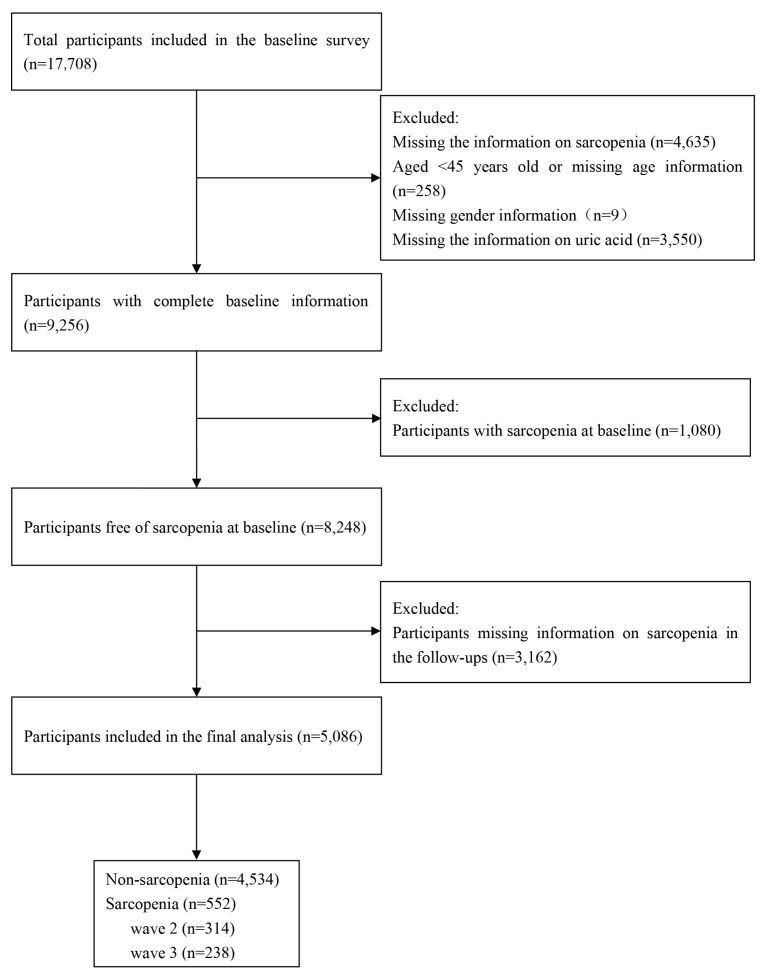
Flowchart of study participants selection.

**Figure 2 nutrients-15-03097-f002:**
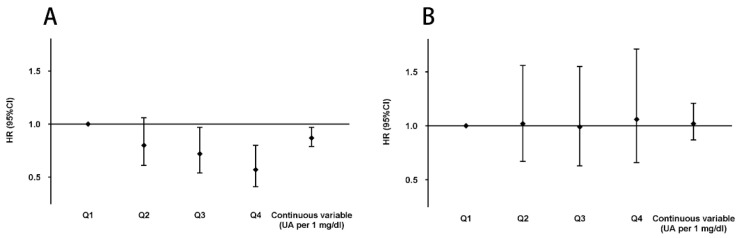
(**A**) HRs and 95% CIs for incident sarcopenia according to the sex-specific quartiles of uric acid and uric acid as a continuous variable in males. (**B**) HRs and 95% CIs for incident sarcopenia according to the sex-specific quartiles of uric acid and uric acid as a continuous variable in females. In males: Q1: ≤4.06 mg/dL; Q2: 4.06–4.80 mg/dL; Q3: 4.81–5.67 mg/dL; Q4: ≥5.67 mg/dL. In females: Q1: ≤3.30 mg/dL; Q2: 3.30–3.88 mg/dL; Q3: 3.88–4.58 mg/dL; Q4: ≥4.58 mg/dL. Adjusted for age, residence, educational level, marital status, drinking status, smoking status, nighttime sleep duration, DM, hypertension, depression, dyslipidemia, pulmonary disease, kidney disease, cardiovascular disease, arthritis, eGFR, CRP, HDL-c and LDL-c. Abbreviations: HRs, hazards ratios; CIs, confidence intervals; UA, uric acid; DM, diabetes mellitus; eGFR, estimated glomerular filtration rate; HDL-c, high-density lipoprotein cholesterol; LDL-c, low-density lipoprotein cholesterol; CRP, high sensitivity C-reactive protein.

**Table 1 nutrients-15-03097-t001:** Baseline characteristics of participants in this study according to sex.

	Total	Male	Female	*p*-Value
*n*	5086	2359	2727	
Uric acid, mg/dL	4.43 ± 1.23	4.93 ± 1.23	4.00 ± 1.05	<0.001
Demographic				
Age, years	58.36 ± 8.34	59.53 ± 8.37	57.34 ± 8.19	<0.001
Residence				0.12
Urban	842 (16.61)	370 (15.74)	472 (17.37)	
Rural	4227 (83.39)	1981 (84.26)	2246 (82.63)	
Education level				<0.001
Primary school or below	3534 (69.48)	1395 (59.14)	2139 (78.44)	
Middle school	1059 (20.82)	640 (27.13)	419 (15.36)	
High school or above	493 (9.69)	324 (13.73)	169 (6.20)	
Marital status				<0.001
Married and living with spouse	4407 (86.65)	2117 (89.74)	2290 (83.98)	
Others	679 (13.35)	242 (10.26)	437 (16.02)	
Health behavior				
Drinking				<0.001
Yes	1663 (32.74)	1333 (56.58)	330 (12.11)	
No	3417 (67.26)	1023 (43.42)	2394 (87.89)	
Smoking				<0.001
Yes	1516 (29.88)	1366 (58.08)	150 (5.51)	
No	3557 (70.12)	986 (41.92)	2571 (94.49)	
Sleep duration, h	6.37 ± 1.82	6.44 ± 1.75	6.31 ± 1.87	0.011
Chronic diseases				
Diabetes Mellitus	750 (14.75)	320 (13.57)	430 (15.77)	0.027
Hypertension	2049 (40.29)	920 (39.00)	1129 (41.40)	0.082
Depression	1623 (31.91)	605 (25.65)	1018 (37.33)	<0.001
Dyslipidemia	1406 (27.64)	620 (26.28)	786 (28.82)	0.043
Pulmonary disease	493 (9.74)	278 (11.86)	215 (7.91)	<0.001
Kidney disease	349 (6.91)	188 (8.04)	161 (5.94)	0.003
Cardiovascular disease	586 (11.59)	241 (10.28)	345 (12.73)	0.007
Arthritis	1788 (35.24)	731 (31.08)	1057 (38.83)	<0.001
Biochemical parameters				
Creatinine, mg/dL	0.77 ± 0.18	0.87 ± 0.18	0.69 ± 0.13	<0.001
eGFR, mL/(min × 1.73 m^2^)	108.37 ± 28.73	104.98 ± 26.70	111.30 ± 30.07	<0.001
HDL-c, mg/dL	50.64 ± 15.07	50.24 ± 16.09	51.00 ± 14.12	0.073
LDL-c, mg/dL	117.19 ± 34.80	112.61 ± 33.68	121.16 ± 35.26	<0.001
CRP, mg/L	2.43 ± 6.60	2.64 ± 7.47	2.25 ± 5.74	0.033
Sarcopenia				
Handgrip strength, kg	33.53 ± 10.15	40.07 ± 8.61	27.88 ± 7.69	<0.001
SMI, kg/m^2^	6.86 ± 1.14	7.69 ± 0.92	6.14 ± 0.78	<0.001
Gait speed, m/s	1.50 ± 4.51	1.56 ± 4.39	1.43 ± 4.63	0.509
5-time chair stand test, s	10.43 ± 3.83	9.92 ± 3.687	10.88 ± 3.90	<0.001
Sarcopenia	552 (10.85)	370 (15.68)	182 (6.67)	<0.001

Continuous variables are presented as the mean ± standard deviation, and categorical variables are expressed as numbers (percentages). Abbreviations: eGFR, estimated glomerular filtration rate; HDL-c, high-density lipoprotein cholesterol; LDL-c, low-density lipoprotein cholesterol; CRP, high sensitivity C-reactive protein; SMI, skeletal muscle mass index.

**Table 2 nutrients-15-03097-t002:** Baseline characteristics of males in this study according to quartiles of uric acid.

	Total	Q1 (≤4.06 mg/dL)	Q2 (4.06–4.80 mg/dL)	Q3 (4.81–5.67 mg/dL)	Q4 (≥5.67 mg/dL)	*p*-Value
*n*	2359	587	589	592	591	
Uric acid, mg/dL	4.93 ± 1.23	3.51 ± 0.45	4.44 ± 0.22 *	5.21 ± 0.25 * ^#^	6.58 ± 0.81 * ^# &^	<0.001
Demographic						
Age, years	59.53 ± 8.37	59.17 ± 7.91	59.27 ± 8.29	59.59 ± 8.56	60.08 ± 8.70	0.236
Residence						<0.001
Urban	370 (15.74)	68 (11.62)	75 (12.78)	105 (17.77) * ^#^	122 (20.75) * ^#^	
Rural	1981 (84.26)	517 (88.38)	512 (87.22)	486 (82.23)	466 (79.25)	
Education level						0.898
Primary school or below	1395 (59.14)	345 (59.25)	349 (59.25)	346 (58.45)	355 (59.14)	
Middle school	640 (27.13)	163 (27.77)	166 (28.18)	161 (27.20)	150 (25.38)	
High school or above	324 (13.73)	79 (13.46)	74 (12.56)	85 (14.36)	86 (14.55)	
Marital status						0.175
Married and living with spouse	2117 (89.74)	519 (88.42)	524 (89.96)	530 (89.53)	544 (92.05)	
Others	242 (10.26)	68 (11.58)	65 (11.04)	62 (10.47)	47(7.95)	
BMI, kg/m^2^	23.56 ± 10.33	22.58 ± 3.66	23.15 ± 3.21	24.41 ± 19.66 *	24.09 ± 3.69	0.008
Health behavior						
Drinking						0.149
Yes	1333 (56.58)	318 (54.27)	322 (54.67)	338 (57.19)	355 (60.17)	
No	1023 (43.42)	268 (45.73)	267 (45.33)	253 (42.81)	235 (39.83)	
Smoking						0.017
Yes	1366 (58.08)	359 (61.37)	361 (61.29)	323 (54.84) * ^#^	323 (54.84) * ^#^	
No	986 (41.92)	226 (38.63)	228 (38.71)	266 (45.16)	266 (45.16)	
Sleep duration, h	6.44 ± 1.75	6.48 ± 1.69	6.42 ± 1.70	6.38 ± 1.80	6.48 ± 1.81	0.686
Chronic diseases						
Diabetes Mellitus	320 (13.57)	85 (14.48)	73 (12.39)	72 (12.16)	90 (15.23)	0.321
Hypertension	920 (39.00)	174 (29.64)	207 (35.14) *	247 (41.72) * ^#^	292 (49.41) * ^# &^	<0.001
Depression	605 (25.65)	193 (32.88)	151 (25.64) *	143 (24.16) *	118 (19.97) * ^# &^	<0.001
Dyslipidemia	620 (26.28)	106(18.06)	143(24.28) *	162(27.36) *	209(35.36) * ^# &^	<0.001
Pulmonary disease	278 (11.86)	77 (13.18)	63 (10.71)	70 (11.63)	68 (11.60)	0.624
Kidney disease	188 (8.04)	48 (8.26)	42 (7.18)	52 (8.86)	46 (7.85)	0.756
Cardiovascular disease	241 (10.28)	51 (8.75)	64 (10.88)	69 (11.75)	57 (9.73)	0.348
Arthritis	731 (31.08)	188 (30.27)	178 (30.27)	187 (31.69)	178 (30.27)	0.864
Biochemical parameters						
Creatinine, mg/dL	0.87 ± 0.18	0.78 ± 0.14	0.84 ± 0.15 *	0.89 ± 0.16 * ^#^	0.97 ± 0.19 * ^# &^	<0.001
eGFR, mL/(min × 1.73 m^2^)	104.9 ± 26.70	119.06 ± 27.67	108.69 ± 24.52	100.91 ± 22.93 * ^#^	91.31 ± 23.39 * ^# &^	<0.001
HDL-c, mg/dL	50.24 ± 16.09	51.02 ± 14.39	51.42 ± 16.89	50.58 ± 16.98 *	47.93 ± 15.98 * ^# &^	<0.001
LDL-c, mg/dL	112.61 ± 33.68	108.19 ± 29.99	113.25 ± 32.19	114.11 ± 32.89 *	114.87 ± 38.71 *	0.002
CRP, mg/L	2.64 ± 7.47	2.68 ± 6.41	2.50 ± 8.65	2.31 ± 5.00	3.09 ± 9.08	0.313
Sarcopenia						
Handgrip strength, kg	40.07 ± 8.61	39.27 ± 8.76	40.04 ± 8.85	40.61 ± 8.66	40.33 ± 8.12	0.047
SMI, kg/m^2^	7.69 ± 0.92	7.53 ± 0.72	7.64 ± 0.64	7.77 ± 1.36 *	7.81 ± 0.74 * ^#^	<0.001
Gait speed, m/s	1.56 ± 4.39	1.42 ± 0.43	1.65 ± 4.62	1.35 ± 0.45	1.80 ± 7.27	0.602
5-time chair stand test, s	9.92 ± 3.68	10.18 ± 3.48	9.92 ± 3.73	9.99 ± 4.03	9.57 ± 3.42 *	0.039
Sarcopenia	370 (15.68)	127 (21.64)	96 (16.30) *	84 (14.19) *	63 (10.66) * ^#^	<0.001

Continuous variables are presented as the mean ± standard deviation, and categorical variables are expressed as numbers (percentages). Abbreviations: BMI, body mass index; eGFR, estimated glomerular filtration rate; HDL-c, high-density lipoprotein cholesterol; LDL-c, low-density lipoprotein cholesterol; CRP, high sensitivity C-reactive protein; SMI, skeletal muscle mass index. Significant at * *p* < 0.05 compared to Q1 group, ^#^
*p* < 0.05 compared to Q2 group, ^&^
*p* < 0.05 compared to Q3 group.

**Table 3 nutrients-15-03097-t003:** Baseline characteristics of females in this study according to quartiles of uric acid.

	Total	Q1 (≤3.30 mg/dL)	Q2 (3.30–3.88 mg/dL)	Q3 (3.88–4.58 mg/dL)	Q4 (≥4.58 mg/dL)	*p*-Value
*n*	2727	681	682	682	682	
Uric acid, mg/dL	4.00 ± 1.05	2.81 ± 0.36	3.59 ± 0.16 *	4.20 ± 0.19 * ^#^	5.41 ± 0.79 * ^# &^	<0.001
Demographic						
Age, years	57.34 ± 8.19	55.68 ± 7.49	57.01 ± 8.02 *	57.33 ± 8.10 *	59.36 ± 8.68 * ^# &^	<0.001
Residence						<0.001
Urban	472 (17.37)	83 (12.21)	97 (14.26)	134 (19.71) * ^#^	158 (23.30) * ^#^	
Rural	2246 (82.63)	597 (87.79)	583 (85.74)	546 (80.29)	520(76.70)	
Education level						0.5
Primary school or below	2139 (78.44)	529 (77.68)	535 (78.45)	539 (79.03)	536 (78.59)	
Middle school	419 (15.36)	115 (16.89)	95(13.93)	105 (15.40)	104 (15.25)	
High school or above	169 (6.20)	37 (5.43)	52 (7.62)	38 (5.57)	42 (6.16)	
Marital status						0.098
Married and living with spouse	2290 (86.20)	587 (86.20)	578 (84.75)	570 (83.58)	555 (81.38)	
Others	437 (16.02)	94 (13.80)	104 (15.25)	112 (16.42)	127 (18.62)	
BMI, kg/m^2^	24.75 ± 9.5	24.59 ± 17.74	24.28 ± 3.89	24.63 ± 3.76	25.49 ± 4.07	0.104
Health behavior						
Drinking						0.294
Yes	330 (12.11)	72 (10.57)	81 (11.88)	82 (12.06)	95 (13.95)	
No	2394 (87.89)	609 (89.43)	601 (88.12)	598 (87.94)	586 (86.05)	
Smoking						0.505
Yes	150 (5.51)	33 (4.85)	41 (6.01)	33 (4.86)	43 (6.33)	
No	2571 (94.49)	648 (95.15)	641 (93.99)	646 (95.14)	636 (93.67)	
Sleep duration, h	6.31 ± 1.87	6.34 ± 1.90	6.29 ± 1.91	6.30 ± 1.83	6.31 ± 1.85	0.966
Chronic diseases						
Diabetes Mellitus	430 (15.77)	105 (15.42)	94 (13.78)	95 (13.93)	136 (19.94) * ^# &^	0.005
Hypertension	1129 (41.40)	226 (33.19)	269 (39.44) *	276 (40.47) *	358 (52.49) * ^# &^	<0.001
Depression	1018 (37.33)	263 (38.62)	256 (37.54)	251 (36.80)	248 (36.36)	0.838
Dyslipidemia	786 (28.82)	141 (20.70)	162 (23.75) *	207 (30.35) * ^#^	276 (40.47) * ^# &^	<0.001
Pulmonary disease	215 (7.91)	49 (7.23)	49 (7.18)	58 (8.55)	59 (8.69)	0.598
Kidney disease	161 (5.94)	41 (6.07)	42 (6.17)	37 (5.47)	41 (6.06)	0.946
Cardiovascular disease	345 (12.73)	77 (11.36)	103 (15.17)	78 (11.50)	87 (12.87)	0.126
Arthritis	1057 (38.83)	233 (34.21)	260 (38.24)	268 (39.41) *	296 (43.47) *	0.006
Biochemical parameters						
Creatinine, mg/dL	0.69 ± 0.13	0.62 ± 0.10	0.67 ± 0.11 *	0.70 ± 0.11 * ^#^	0.77 ± 0.15 * ^# &^	<0.001
eGFR, mL/(min × 1.73 m^2^)	111.30 ± 30.07	125.99 ± 31.45	114.37 ± 26.04 *	108.55 ± 28.32 * ^#^	96.29 ± 26.31 * ^# &^	<0.001
HDL-c, mg/dL	51.00 ± 14.12	52.74 ± 14.13	52.44 ± 14.34	50.37 ± 13.65 * ^#^	48.43 ± 13.96 * ^# &^	<0.001
LDL-c, mg/dL	121.16 ± 35.26	117.02 ± 33.09	118.66 ± 32.22	122.98 ± 35.82 * ^#^	126.01 ± 38.92 * ^#^	<0.001
CRP, mg/L	2.25 ± 5.74	1.83 ± 5.52	2.18 ± 6.70 *	2.11 ± 4.39 * ^#^	2.88 ± 6.04 * ^# &^	0.006
Sarcopenia						
Handgrip strength, kg	27.88 ± 7.69	27.94 ± 7.08	27.91 ± 8.44	27.71 ± 8.16	27.97 ± 7.69	0.922
SMI, kg/m^2^	6.14 ± 0.78	6.07 ± 0.81	6.07 ± 0.72	6.15 ± 0.74	6.28 ± 0.82 * ^#^	<0.001
Gait speed, m/s	1.43 ± 4.63	1.34 ± 0.67	1.32 ± 0.38	1.26 ± 0.40	1.71 ± 8.37	0.644
5-time chair stand test, s	10.87 ± 3.90	10.75 ± 3.34	10.80 ± 3.58	10.94 ± 3.97	11.00 ± 4.60	0.624
Sarcopenia	182 (6.67)	46 (6.75)	49 (7.18)	44 (6.45)	43 (6.30)	0.92

Continuous variables are presented as the mean ± standard deviation, and categorical variables are expressed as numbers (percentages). Abbreviations: BMI, body mass index; eGFR, estimated glomerular filtration rate; HDL-c, high-density lipoprotein cholesterol; LDL-c, low-density lipoprotein cholesterol; CRP, high sensitivity C-reactive protein; SMI, skeletal muscle mass index. Significant at * *p* < 0.05 compared to Q1 group, ^#^
*p* < 0.05 compared to Q2 group, ^&^
*p* < 0.05 compared to Q3 group.

**Table 4 nutrients-15-03097-t004:** Hazard ratios and 95% confidence intervals for incident sarcopenia according to the sex-specific quartiles of uric acid and uric acid as a continuous variable.

Outcome	Cases	Incidence Rate, per 1000 Person-Years	Model 1	*p*-Value	Model 2	*p*-Value	Model 3	*p*-Value	Model 4	*p*-Value
Males										
Continuous variable (UA per 1 mg/dL)	370	42.47	0.81 (0.74–0.88)	<0.001	0.81 (0.74–0.89)	<0.001	0.86 (0.79–0.95)	0.002	0.87 (0.79–0.97)	0.011
UA quartiles										
Q1 (≤4.06 mg/dL)	127	60.13	1		1		1		1	
Q2 (4.06–4.80 mg/dL)	96	44.24	0.73 (0.56–0.95)	0.023	0.71 (0.55–0.93)	0.015	0.81 (0.62–1.06)	0.127	0.80 (0.61–1.06)	0.124
Q3 (4.81–5.67 mg/dL)	84	38.18	0.62 (0.47–0.82)	0.001	0.64 (0.48–0.84)	0.002	0.73 (0.55–0.97)	0.029	0.72 (0.54–0.97)	0.036
Q4 (≥5.67 mg/dL)	63	28.27	0.44 (0.32–0.60)	<0.001	0.45 (0.33–0.61)	<0.001	0.56 (0.41–0.76)	<0.001	0.57 (0.41–0.80)	0.001
Females										
Continuous variable (UA per 1 mg/dL)	182	17.11	0.88 (0.76–1.01)	0.091	0.88 (0.75–1.02)	0.089	0.92 (0.79–1.07)	0.308	1.02 (0.87–1.21)	0.74
UA quartiles							1		1	
Q1 (≤3.30 mg/dL)	46	17.26	1		1					
Q2 (3.30–3.88 mg/dL)	49	18.46	0.96 (0.64–1.43)	0.851	0.90 (0.60–1.36)	0.641	0.91 (0.60–1.38)	0.676	1.02 (0.67–1.56)	0.911
Q3 (3.88–4.58 mg/dL)	44	16.48	0.84 (0.55–1.27)	0.423	0.81 (0.53–1.24)	0.34	0.82 (0.53–1.26)	0.369	0.99 (0.63–1.55)	0.983
Q4 (≥4.58 mg/dL)	43	16.21	0.72 (0.47–1.10)	0.13	0.72 (0.47–1.11)	0.139	0.81 (0.52–1.26)	0.355	1.06 (0.66–1.71)	0.796

Data are presented as HRs (95%CIs). Model 1: adjusted age, residence, educational level, marital status. Model 2: further adjusted drinking status, smoking status, nighttime sleep duration based on model 1. Model 3: further adjusted DM, hypertension, depression, dyslipidemia, pulmonary disease, kidney disease, cardiovascular disease, arthritis based on model 2. Model 4: further adjusted eGFR, CRP, HDL-c and LDL-c based on model 3. Abbreviations: HRs, hazards ratios; CIs, confidence intervals; UA, uric acid; DM, diabetes mellitus; eGFR, estimated glomerular filtration rate; HDL C, high-density lipoprotein cholesterol; LDL-c, low-density lipoprotein cholesterol; CRP, high sensitivity C-reactive protein.

**Table 5 nutrients-15-03097-t005:** Hazard ratios and 95% confidence intervals for incident sarcopenia in subgroup analyses.

	HRs (95%CIs) ^a^	
	Males	Females
Age		
≤58 years old	0.73 (0.58–0.92)	1.14 (0.82–1.59)
>58	0.92 (0.82–1.03)	1.05 (0.87–1.26)
Residence		
Urban	1.03 (0.73–1.47)	1.54 (0.93–2.55)
Rural	0.86 (0.77–0.95)	1.00 (0.84–1.20)
Smoking		
Yes	0.88 (0.77–0.99)	1.10 (0.70–1.73)
No	0.88 (0.73–1.06)	1.00 (0.84–1.20)
Drinking		
Yes	0.90 (0.77–1.05)	0.88 (0.55–1.41)
No	0.85 (0.74–0.98)	1.03 (0.86–1.23)
Hypertension		
Yes	0.93 (0.77–1.12)	1.05 (0.82–1.34)
No	0.80 (0.76–0.97)	1.02 (0.81–1.28)
Diabetes Mellitus		
Yes	1.36 (0.92–2.01)	1.08 (0.68–1.71)
No	0.85 (0.76–0.94)	1.03 (0.87–1.23)
Dyslipidemia		
Yes	0.96 (0.68–1.36)	0.72 (0.49–1.05)
No	0.87 (0.79–0.97)	1.16 (0.97–1.39)
Depression		
Yes	0.78 (0.64–0.94)	1.08 (0.85–1.36)
No	0.91 (0.80–1.03)	0.98 (0.77–1.24)

Data are presented as HRs (95%CIs). ^a^ Adjusted for age, residence, educational level, marital status, drinking status, smoking status, nighttime sleep duration, DM, hypertension, depression, dyslipidemia, pulmonary disease, kidney disease, cardiovascular disease, arthritis, eGFR, CRP, HDL-c and LDL-c. Abbreviations: HRs, hazards ratios; CIs, confidence intervals; eGFR, estimated glomerular filtration rate; HDL-c, high-density lipoprotein cholesterol; LDL-c, low-density lipoprotein cholesterol; CRP, high sensitivity C-reactive protein.

## Data Availability

The datasets of our study are from the CHARLS website: http://charls.pku.edu.cn/en (accessed on 8 June 2023).

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
