# Peer review of "Longitudinal Association between Uric Acid and Incident Sarcopenia"

_nutrients, 2023, doi:10.3390/nu15143097_

Round 1
Reviewer 1 Report
Zhou et al. investigate the association between uric acid and sarcopenia.
My main concerns are the collection of data and the way it is presented/described. Repeatedly, in the article, it is emphasized that the novelty of the paper is its "longitudinal" but from the contents, it appears that the only characteristic observed years later is the absence/presence of sarcopenia. There is no information on whether the baseline characteristic came from waves 1, 2, or 3. There is no information on what period after wave 1 the diagnosis was made.
Moreover, it is known that coffee consumption is also associated with sarcopenia, were such data collected?
The authors in the discussion emphasize that nutritional status is important, but such data were not collected, nor was the BMI of the people included in the analysis shown
My specific comments are listed below:
- Line 19: “15.68%” - it is not clear what this percentage is. of all participants? all patients? All men?
- Line 37: some modifiable risk factors are already known, it should be noted
- Line 130: Were any pots-hoc tests performed?
- Line 130: How assumptions of ANOVA was checked?
- Figures and tables are far after their appearance in the main text
- The table header should be on the same page as the table
- Line 227: “According…..” - it seems there is no end to this sentence
- Tables - The p-value would be more informative if it specified between which values the difference is statistically significant
- Tables 2, 3, 4 - whenever it is possible, the tables should be placed on one page
- Line 263: even in the case of drinking and dyslipidemia?
- Line 281/284: in male? Female? Both?
- Line 290: Why are different provinces important in terms of UA level?
- Line 302: reference refers to hemodialysis patients so conclusions cannot be extended to the general population
- Line 308: was information about menopause and hormone replacement therapy collected?
- Supplementary materials section was not completed
- Authors contribution: no information if all authors have read and agreed to the published version of the manuscript
Author Response
Response to Reviewer 1 Comments
Comments and Suggestions for Authors
Zhou et al. investigate the association between uric acid and sarcopenia.
- My main concerns are the collection of data and the way it is presented/described. Repeatedly, in the article, it is emphasized that the novelty of the paper is its "longitudinal" but from the contents, it appears that the only characteristic observed years later is the absence/presence of sarcopenia. There is no information on whether the baseline characteristic came from waves 1, 2, or 3. There is no information on what period after wave 1 the diagnosis was made.
Reply:Thanks very much for your comment and suggestion. The baseline characteristic came from wave 1. We have changed the flowchart to show the number of incident sarcopenia that diagnosed in wave 2 and wave 3. A total of 552 patients diagnosed as sarcopenia in the follow ups, and 314 were from wave 2, and 238 were from wave 3.
- Moreover, it is known that coffee consumption is also associated with sarcopenia, were such data collected?
Reply:Thanks very much for your comment and suggestion. The information on the coffee consumption or the data of dietary intake were not collected in the CHARLS. The included participants in this study were mostly from the rural regions of China. The number of patients with the coffee consumption habit may be limited. But we thank for you advice sincerely, and discussed this in the part of limitation.
- The authors in the discussion emphasize that nutritional status is important, but such data were not collected, nor was the BMI of the people included in the analysis shown
Reply:Thanks very much for your comment and suggestion. (1) The data of diet and nutritional status were not collected in the CHARLS. This is one of the limitations of our study and we have discussed in the discussion. (2) We have compared the baseline BMI of participants according to the quartiles of uric acid and the results were shown in the Table 2 and Table 3. We did not find any significant difference in females. In males, the BMI showed significant difference between Q3 and Q1 uric acid groups.
My specific comments are listed below:
- Line 19: “15.68%” - it is not clear what this percentage is. of all participants? all patients? All men?
Reply:Thanks very much for your comment and suggestion. To avoid causing confusion again, we have deleted the percentage. This percentage means the ratio patients with sarcopenia to all included men.
- Line 37: some modifiable risk factors are already known, it should be noted
Reply:Thanks very much for your comment and suggestion. We have added this sentence to the end of this paragraph: “Physical inactivity and poor diet have been reported to be risk factors of sarcopenia, but the association between uric acid and sarcopenia remained unclear.”
- Line 130: Were any pots-hoc tests performed?
Reply:Thanks very much for your comment and suggestion. We have added post-hoc tests to make the P-values to be more informative, and the results were shown in Table 2 and Table 3. For continuous variables with P-value <0.05, the post-hoc analyses were conducted by the Scheffe’s method.
- Line 130: How assumptions of ANOVA was checked?
Reply:Thanks very much for your comment and suggestion. The normality of continuous variables reported in our study is inspected visually by QQ plot. The Bartlett’s test was used to test homoscedasticity.
- Figures and tables are far after their appearance in the main text
Reply:Thanks very much for your comment and suggestion. We have changed the position of the tables and figures.
- The table header should be on the same page as the table
Reply:Thanks very much for your comment and suggestion. We have put the table leaders on the same page as the tables.
- Line 227: “According…..” - it seems there is no end to this sentence
Reply:Thanks very much for your comment and suggestion. We have changed the sentence to “ According to the subgroup analyses, males who were younger (≤ 58 years old), living in rural, smokers, non-drinkers, and without hypertension, DM, dyslipidemia, as well as with depression were more responsive to the protective effects of UA against the incident of sarcopenia”
- Tables - The p-value would be more informative if it specified between which values the difference is statistically significant?
Reply:Thanks very much for your comment and suggestion. We have added post-hoc tests to make the P-values to be more informative, and the results were shown in Table 2 and Table 3. For continuous variables with P-value <0.05, the post-hoc analyses were conducted by the Scheffe’s method.
- Tables 2, 3, 4 - whenever it is possible, the tables should be placed on one page
Reply:Thanks very much for your comment and suggestion. We failed to place these tables on one page.
- Line 263: even in the case of drinking and dyslipidemia?
Reply:Thanks very much for your comment and suggestion. In the subgroup analyses, males who were non-drinkers and who were without dyslipidemia were found to be more responsive to the protective effects of uric acid against the incident sarcopenia (Table 5). In males with alcohol consumption and dyslipidemia, we did not find significant association between uric acid and sarcopenia.
- Line 281/284: in male? Female? Both?
Reply:Thanks very much for your comment and suggestion. Our findings indicated a significant negative association between uric acid and incident sarcopenia in males. In females, we did not find any significant association between uric acid and sarcopenia.
- Line 290: Why are different provinces important in terms of UA level?
Reply:Thanks very much for your comment and suggestion. Participants in the study conducted in West China were only collected from four provinces. But we used the data of CHARLS, which collected participants from 450 villages or residential communities in 28 provinces throughout China. Thus, the discrepancy of results may make sense.
- Line 302: reference refers to hemodialysis patients so conclusions cannot be extended to the general population
Reply:Thanks very much for your comment and suggestion. We agreed to your suggestion, and changed it into “This study revealed that the UA was positively associated with relative ASM in patients with peritoneal dialysis and this association only existed in males. However, the study population was limited to peritoneal dialysis patients, and further studies involving general population are needed”.
- Line 308: was information about menopause and hormone replacement therapy collected?
Reply:Thanks very much for your comment and suggestion. The information about menopause and hormone replacement therapy were not collected in CHARLS, and the data were not available.
- Supplementary materials section was not completed
Reply:Thanks very much for your comment and suggestion. We have completed the supplementary materials section.
- Authors contribution: no information if all authors have read and agreed to the published version of the manuscript
Reply:Thanks very much for your comment and suggestion. We have added “All authors have read and agreed to the published version of the manuscript” to the part of Author Contributions.

Reviewer 2 Report
Review for the manuscript
LONGITUDINAL ASSOCIATION BETWEEN URIC ACID AND INCIDENT 2 SARCOPENIA
Dear Editor and authors,
Thank you for the opportunity to review this interesting manuscript. I have some suggestions before it can be considered for publication in Nutrients.
TITLE
It is adequate.
ABSTRACT
In line 15-16, the authors say that “Due to the sex differences in UA, the UA levels were analyzed as sex-specific quartiles and also as a 15 continuous variable (per 1mg/dl).” Please, re-formulate this sentence and change mg/dl for mg/dL (do that along all the text)…
KEYWORDS
Please, include at least one more key-word.
INTRODUCTION
This section is adequate, however, I suggest including newer references. At PUBMED the authors can find a plethora of articles published in 2022 and 2023 related to sarcopenia.
METHODS
Please, define “CHARLS”.
In line 60-63 we can read: “The CHARLS employed a multistage probability proportional to size sampling strategy to select households from 450 villages or residential communities in 28 provinces throughout China All recruited participants were interviewed through a standardized face-to-face computer-assisted personal interview (CAPI) by well-trained interviewers…”. I suggest changing for: “The CHARLS employed a multistage probability proportional to size sampling strategy to select households from 450 villages or residential communities in 28 provinces throughout China. All recruited participants were interviewed through standardized face-to-face computer-assisted personal interviews (CAPI) by well-trained interviewers…”.
In line 71 we can see uric acid, and previously the authors used UA.
In lines 75-76, please include de data of the Ethics Committee approval.
I suggest including a flowchart showing each wave and the number of patients included to facilitate visualization (improve Figure 1 or build another).
RESULTS
Please, improve the quality of Figures 2 A and 2B. If the authors maintain these figures separated, as we see in the text, I suggest naming Figure 2 (for Figure 2A) and Figure 3 for Figure 2B. However, if the authors join these Figures, they could use only one legend, and this would be easier to readers.
Please, change mg/dl for mg/dL; HDL-C for HDL-c, LDL-C for LDL-c, and m2 for m2, along with the entire text.
In table 1 the authors use “HDL-c” and “LDL-c”. In table 2 they use only “HDL” and “LDL”. The same is true for table 3.
In line 267 we see: “…association between UA and sarcopenia did not change significantly (Table A2-3).” What is Table A2-3? It is not in the text.
DISCUSSION
In lines 320-322 we can read: “First, the primary limitation was that the muscle mass was estimated by a formula, not by DXA or bioelectrical impedance analysis. But the formula has been previously validated in Chinese population and showed good consistency with DXA in various studies [18, 19].” I suggest changing for: “First, although the formula has been previously validated in Chinese population and showed good consistency with DXA in various studies [18, 19], the muscle mass was estimated by a formula, not by DXA or bioelectrical impedance analysis…”
CONCLUSION
Please, expand this section.
REFERENCES
As suggested before, please, include more references published in 2022 and 2023 in the Introduction and Discussion sections.
Minor corrections are necessary.
Author Response
Response to Reviewer 2 Comments
TITLE
- It is adequate.
Reply:Thanks very much for your comment and suggestion.
ABSTRACT
- In line 15-16, the authors say that “Due to the sex differences in UA, the UA levels were analyzed as sex-specific quartiles and also as a 15 continuous variable (per 1mg/dl).” Please, re-formulate this sentence and change mg/dl for mg/dL (do that along all the text)…
Reply:Thanks very much for your comment and suggestion. We have changed this sentence into “Due to the sex differences, the UA levels were analyzed by categorizing into sex-specific quartiles or by using UA levels as a continuous variable (per 1mg/dL)”.
KEYWORDS
- Please, include at least one more key-word.
Reply:Thanks very much for your comment and suggestion. We have changed the key words into “sarcopenia; uric acid; longitudinal study; CHARLS”.
INTRODUCTION
- This section is adequate, however, I suggest including newer references. At PUBMED the authors can find a plethora of articles published in 2022 and 2023 related to sarcopenia.
Reply:Thanks very much for your comment and suggestion. We have added some new articles in the introduction and discussion section as references.
METHODS
- Please, define “CHARLS”.
Reply:Thanks very much for your comment and suggestion. We have defined “CHARLS”. The full name of the CHARLS first appeared in the end of the introduction: “Therefore, to better understand the effects of UA on sarcopenia, we conducted this study to explore the longitudinal association between UA and incident sarcopenia among the middle-aged and older adults based on the nationally representative data from the China Health and Retirement Longitudinal Study (CHARLS)”. The definition of CHARLS was in the method section.
- In line 60-63 we can read: “The CHARLS employed a multistage probability proportional to size sampling strategy to select households from 450 villages or residential communities in 28 provinces throughout China All recruited participants were interviewed through a standardized face-to-face computer-assisted personal interview (CAPI) by well-trained interviewers…”. I suggest changing for: “The CHARLS employed a multistage probability proportional to size sampling strategy to select households from 450 villages or residential communities in 28 provinces throughout China. All recruited participants were interviewed through standardized face-to-face computer-assisted personal interviews (CAPI) by well-trained interviewers…”.
Reply:Thanks very much for your comment and suggestion. We have changed the sentence.
- In line 71 we can see uric acid, and previously the authors used UA.
Reply:Thanks very much for your comment and suggestion. We have changed into UA.
- In lines 75-76, please include de data of the Ethics Committee approval.
Reply:Thanks very much for your comment and suggestion. The Ethical approval for the CHARLS was obtained from the Institutional Review Board of Peking University (approval number: IRB00001052–11,015).
- I suggest including a flowchart showing each wave and the number of patients included to facilitate visualization (improve Figure 1 or build another).
Reply:Thanks very much for your comment and suggestion. We have changed the flowchart to show the number of incident sarcopenia that diagnosed in each wave. A total of 552 patients diagnosed as sarcopenia in the follow ups, and 314 were from wave 2, and 238 were from wave 3.
RESULTS
- Please, improve the quality of Figures 2 A and 2B. If the authors maintain these figures separated, as we see in the text, I suggest naming Figure 2 (for Figure 2A) and Figure 3 for Figure 2B. However, if the authors join these Figures, they could use only one legend, and this would be easier to readers.
Reply:Thanks very much for your comment and suggestion. We have jointed the two figures.
- Please, change mg/dl for mg/dL; HDL-C for HDL-c, LDL-C for LDL-c, and m2 for m2, along with the entire text.
Reply:Thanks very much for your comment and suggestion. We have carefully checked our manuscript and changed the mistakes.
- In table 1 the authors use “HDL-c” and “LDL-c”. In table 2 they use only “HDL” and “LDL”. The same is true for table 3.
Reply:Thanks very much for your comment and suggestion. We have carefully checked the tables and changed the mistakes.
- In line 267 we see: “…association between UA and sarcopenia did not change significantly (Table A2-3).” What is Table A2-3? It is not in the text.
Reply:Thanks very much for your comment and suggestion. We have changed the Table A2-3 to Table S2-3. These tables were included in the Supplementary materials.
DISCUSSION
- In lines 320-322 we can read: “First, the primary limitation was that the muscle mass was estimated by a formula, not by DXA or bioelectrical impedance analysis. But the formula has been previously validated in Chinese population and showed good consistency with DXA in various studies [18, 19].” I suggest changing for: “First, although the formula has been previously validated in Chinese population and showed good consistency with DXA in various studies [18, 19], the muscle mass was estimated by a formula, not by DXA or bioelectrical impedance analysis…”
Reply:Thanks very much for your comment and suggestion. We have changed this sentence.
CONCLUSION
- Please, expand this section.
Reply:Thanks very much for your comment and suggestion. We have expanded the conclusion to “In conclusion, our findings indicated that UA was negatively associated with incident sarcopenia in males but not in females in middle-aged and older Chinese. The findings supported the association between UA and sarcopenia, and suggested the sex difference existed in the effects of UA on sarcopenia”.
REFERENCES
- As suggested before, please, include more references published in 2022 and 2023 in the Introduction and Discussion sections.
Reply:Thanks very much for your comment and suggestion. We have added some new articles in the introduction and discussion section as references.

Round 2
Reviewer 1 Report
I have no further comments